# Primary Amine Nucleophilic Addition to Nitrilium *Closo*-Dodecaborate [B_12_H_11_NCCH_3_]^−^: A Simple and Effective Route to the New BNCT Drug Design

**DOI:** 10.3390/ijms222413391

**Published:** 2021-12-13

**Authors:** Alexey V. Nelyubin, Nikita A. Selivanov, Alexander Yu. Bykov, Ilya N. Klyukin, Alexander S. Novikov, Andrey P. Zhdanov, Natalia Yu. Karpechenko, Mikhail S. Grigoriev, Konstantin Yu. Zhizhin, Nikolay T. Kuznetsov

**Affiliations:** 1Kurnakov Institute of General and Inorganic Chemistry, Russian Academy of Sciences, Leninskii pr. 31, 119991 Moscow, Russia; nelyubin.av@yandex.ru (A.V.N.); GooVee@yandex.ru (N.A.S.); bykov@igic.ras.ru (A.Y.B.); klukinil@igic.ras.ru (I.N.K.); zhizhin@igic.ras.ru (K.Y.Z.); boron@igic.ras.ru (N.T.K.); 2Institute of Chemistry, Saint Petersburg State University, Universitetskaya Nab. 7-9, 199034 Saint Petersburg, Russia; ja2-88@mail.ru; 3N. N. Blokhin National Medical Research Center of Oncology, 24 Kashirskoye Shosse, 115478 Moscow, Russia; nojadg@mail.ru; 4Depatment of Medicinal Chemistry and Toxicology, Pirogov Russian National Research Medical University, 117997 Moscow, Russia; 5Frumkin Institute of Physical Chemistry and Electrochemistry, Russian Academy of Sciences, Leninskii pr. 31, Bldg 4, 119071 Moscow, Russia; mickgrig@mail.ru

**Keywords:** *closo*-dodecaborate, nitrilium derivative, amines, nucleophilic addition, reaction mechanism

## Abstract

In the present work, a convenient and straightforward approach to the preparation of borylated amidines based on the *closo*-dodecaborate anion [B_12_H_11_NCCH_3_NHR]−, R=H, Alk, Ar was developed. This method has two stages. A nitrile derivative of the general form [B_12_H_11_NCCH_3_]^−^ was obtained, using a modified technique, in the first stage. On the second stage the resulting molecular system interacted with primary amines to form the target amidine products. This approach is characterised by a simple chemical apparatus, mild conditions and high yields of the final products. The mechanism of the addition of amine to the nitrile derivative of the *closo*-dodecaborate anion was studied, using quantum-chemical methods. The interaction between NH_3_ and [B_12_H_11_NCCH_3_]^−^ ammonia was chosen as an example. It was found that the structure of the transition state determines the stereo-selectivity of the process. A study of the biological properties of borylated amidine sodium salts indicated that the substances had low toxicity and could accumulate in cancer cells in significant amounts.

## 1. Introduction

Organic nitriles RCN are excellent starting platforms (precursors) for preparing various valuable organic compounds, such as amides, amidines, N-heterocycles, etc. [1,2,3,4]. The main approach to functionalising this class of compounds is nucleophilic addition to nitrile group CN [5,6,7]. Nevertheless, this type of reaction has a significant activation barrier, which obstructs the preparation of target products. The solution to this problem is the activation of CN group in RCN molecules [8,9]. The initial approach to this problem was based on using nitriles with an electron-accepting R-group, but this approach proved to be insufficient, even in the case of a strong acceptor group such as -CCl_3_.

An alternative approach is based on the initial complexation of organonitriles with transition metals [10,11,12,13]. In complexes where nitriles are coordinated to metal atoms, the CN group is more active compared to initial nitriles [14,15,16,17]. The addition process to transition metal nitrile complexes with C-, N- and O-nucleophiles occurred in mild conditions with high yields [18,19]. Depending on the type of metal, these complexes can be stable, and process of nitrile fragment substitution to another ligand is not performed. It is noteworthy that not only do nitrile complexes attract attention as precursors in organic synthesis, but these substances and their derivatives also have many potential applications: catalysis, photoluminescence and enzymatic transformations [20,21,22,23].

One of the branches of nitrile complexes is molecular systems, where nitrogen atoms are attached to one atom of the boron cluster (*closo*-borate, carborane, bis-dicarbollides, etc.) [24,25,26]. It is noteworthy that the boron clusters and their derivatives are versatile building blocks for formation platforms with plenty of useful properties [27,28]. These molecular systems are applied in different fields, such as medicine [29,30,31,32,33], catalysis [34,35] and electrochemistry, as solid-state battery components [36,37,38] etc.

The preparation process for borylated nitriles is based on electrophile induced nucleophilic substitution (EINS). Bronsted and Lewis acids may act as electrophile inductors. Depending on the type of inductor and its amount in reactive mixture, it is possible to vary the degree of substitution in boron clusters [39]. The chemistry of borylated nitriles is based on the main concepts and approaches of nitrile complex chemistry. The main advantage is the increased stability of B-N bonds. These molecular species are not associated with nucleophilic substitution with the breaking of B-N bonds. This makes it possible to obtain boron clusters with different *exo*-polyhedral substituents, such as amidines, amides, etc. As in the case of nitrile complexes, C-, N- and O-nucleophiles can act as nucleophiles. One of the most exciting features of borylated nitriles, also known as nitrile complexes, is the possibility of the formation of different stereo-isomers: E and Z. The main driving force behind the formation of E- or Z-isomers is non-covalent interactions. If a nucleophile has a polarised hydrogen atom, the dihydrogen bond between the hydride atom of the boron cage and the proton from the nucleophile is formed, and a Z-isomer is obtained.

As in the case of complexes with metals, borylated nitriles can be involved in nucleophilic addition reactions. Similar to nitrile complexes, processes of nucleophilic addition take place in mild conditions with high yields. Collaboration with Professor Kukushkin’s group has involved the investigation of the reactivity of nitrile derivatives of *closo*-decaborate anions with different nucleophiles [40,41,42,43]. The process of alcohol, amine and oxime addition to nitrilium derivatives has been investigated previously [44,45,46,47,48] and the main stereochemical, thermodynamic and kinetic parameters have been established.

Thus, in the present work, the main approach of nitrile complex and borylated nitrile chemistry has been expanded and an investigation has been made of the synthesis and properties of molecular systems based on the *closo*-dodecaborate anion. The chemistry of the *closo*-dodecaborate anion is currently being widely studied [49,50,51,52]. The dodecaborate anion [B_12_H_12_]^2−^ is the most stable anion among all the *closo*-borate families and is neither oxidised nor subject to the opening reaction of *closo*-cage in the presence of strong electrophiles. Derivatives with different *exo*-polyhedral substituents, such as oxonium, ammonium and the carbonyl group, have been obtained [53,54,55,56,57]. In addition, the chemistry of closo-dodecaborate derivatives with *exo*-polyhedral B-N bonds are studied actively [58,59,60]. In most cases anion [B_12_H_11_NH_3_]^−^ was used as starting building block for following functionalization [61,62,63].

Previously, the authors discovered an approach for preparing *closo*-dodecaborate borylated nitriles based on the microwave-assisted interaction of [B_12_H_12_]^2−^ with organic nitriles in the presence of CF_3_COOH [64]. In the present work, this method has been improved. The reactivity of [B_12_H_11_NCCH_3_]^−^ with primary amines has been explored and the crystal structures of some products have been investigated. Some parameters of the biological activity of the products obtained have been established and theoretical modelling of the reaction mechanism has been carried out. The main driving force of the nucleophilic addition process to [B_12_H_11_NCCH_3_]^−^ has been explored theoretically and experimentally.

## 2. Results

### 2.1. Synthesis of Nitrilium Derivatives of the Closo-Dodecaborate Anion

The first task of the current project was the optimisation of the preparation procedure of the initial nitrilium derivatives of *closo*-dodecaborate anions with the general form [B_12_H_11_NCCH_3_]^−^. As mentioned above, nitrilium derivatives are suitable starting platforms for the preparation of various borylated compounds. Thus, finding a straightforward and practical synthetic protocol is an important task. 

Previously, the authors have reported on obtaining acetonitrile derivatives of *closo*-dodecaborate anions using microwave synthesis. It is noteworthy that the formation of the target product does not proceed if the reaction takes place at the boiling temperature of acetonitrile. The application of microwave synthesis allows one to obtain the desired derivatives with high yields. The main disadvantage of the previously reported method is the necessity for complex equipment when applying microwave synthesis. Carrying out the reaction in a glass pressure vessel at 150 °C (temperature of oil bath) precludes the need for expensive equipment (Figure 1). It was found that, at this temperature, the complete conversion of the initial *closo*-dodecaborate anion occurs in about 30 min. Increasing the reaction time has a negative effect on the yield of the target nitrile derivative.

The process of amine addition has an electrophile induced nucleophilic substitution (EINS) mechanism. The choice of an electrophilic inductor significantly affects the composition of the obtained products. The use of various organic acids such as trifluoroacetic acid, *p*-toluenesulfonic acid and trifluoromethanesulfonic acid as inducers proved to be a good decision for the preparation of nitrile derivatives of the *closo*-decaborate anion. In the present research, it was found that, in the case of the [B_12_H_12_]^2−^ anion, all acids except trifluoroacetic acid led to the formation of products of the general type [B_12_H_11_OC(O)R]^2−^ or [B_12_H_11_OSO_2_R]^2−^.

In contrast to the process for the preparation of nitrilium derivatives of the *closo*-decaborate anion, this reaction is sensitive to the presence of traces of organochlorine solvents such as dichloromethane (DCM). Another significant limitation is the necessity to use freshly distilled trifluoroacetic acid. It was found that the use of commercial trifluoroacetic acid produces a by-product with an unknown structure, which is not removed during the extraction of the target nitrile derivative. The nature of the cation used does not affect the composition of the final products. The completeness of the reaction was monitored using ^11^B NMR. Two signals were observed in the spectra of the nitrilium derivative obtained [B_12_H_11_NCCH_3_]^−^. a signal from the boron atom attached to the nitrilium substituent at −13.5 ppm and one from unsubstituted boron atoms B(2–12) at −16.3 ppm. The spectra of the compounds obtained did not depend on the nature of the cation but provided information about the composition of the obtained products.

The nitrilium derivatives obtained were characterised using ^1^H and ^13^C NMR, IR and ESI MS methods. In ^1^H NMR spectra, signals from the tetrabutylammonium cation were observed: multiplets at 3.15, 1.61, 1.45 and 1.01 ppm, and a signal from the protons of the methyl group of the substituent at 2.61 ppm. 

In ^13^C NMR spectra, tetrabutylammonium cation signals were observed at 59.4, 24.4, 20.2 and 13.9 ppm for (NBu4)^+^. Two signals represented the nitrilium substituent: the carbon atom attached to nitrogen at 108.9 ppm and the methyl group’s carbon atom at 4.5 ppm. It is interesting to compare the chemical shift of nitrilium group carbon atoms in various related compounds. In the case of [2-B_10_H_9_NCCH_3_]^−^, the signal of the C-atom of the cyano-group was found at 115.4 ppm. Therefore, it can be concluded that the *closo*-dodecaborate anion had a greater donating effect than the *closo*-decaborate anion.

In the IR-spectra of [B_12_H_11_NCCH_3_]^−^, there are two most significant absorption bands: 2500 cm^−1^ (BH) and 2352 cm^−1^ (C≡N). The value of the absorption band of the C≡N-group of [B_12_H_11_NCCH_3_]^−^ is quite similar to that of [2-B_10_H_9_NCCH_3_]^−^ (2348 cm^−1^). This indicates that C≡N bond orders for two cluster species were fairly similar.

Thus, the method for the preparation of nitrilium derivatives of *closo*-dodecaborate anions has been improved. Applying the glass pressure vessel technique makes it possible to obtain goal products easily and conveniently. High yields of desired products characterise this process. This method is suitable for the preparation of nitrilium derivatives with alkyl and aromatic side chains.

This section may be divided by subheadings. It should provide a concise and precise description of the experimental results, their interpretation, as well as the experimental conclusions that can be drawn.

### 2.2. Interaction with Primary Amines

At the next stage, the addition reactions of primary amines to nitrilium derivatives of *closo*-dodecaborate anions were explored. Amines with different structures were chosen to investigate how the nature of amines influences the addition reaction process. A similar investigation had been conducted on the nitrilium derivatives of *closo*-decaborate anions [65]. In the case of the [B_10_H_10_]^2−^ anion, this process is characterised by mild reaction conditions and high yields. Thus, in the present study, a similar investigation was conducted in the context of *closo*-dodecaborate anions.

The general reaction scheme can be described as follows (Figure 2). Dichloromethane and tetrahydrofuran were chosen as solvents and the addition reaction was completed in 15 min at room temperature. In the case of ammonia and CH_3_NH_2_, reactions took place in water/ethanol media. In such cases, reaction mixtures were refluxed due to the low solubility of [B_12_H_11_NCCH_3_]^−^ derivatives in water/ethanol solution. It is noteworthy that no side-product of the addition process of water/ethanol was obtained. In addition, the interaction process with a bi-functional nucleophile such as monoethanolamine was highly regio-selective. Electrophilic attack of the nitrilium group was performed only on the amino group of monoethanolamine, as shown by the IR and ^1^H NMR spectra of the resulting product. Obviously, the addition process of N-nucleophiles predominates over the analogue process with O-nucleophiles, and this has many advantages. For example, it is possible for the reaction to take place in water solvent, without using organic solvents.

The completeness of the reaction was monitored using ^11^B NMR. In the spectra of the amidines signal obtained, the substituted boron atom was shifted to the low frequency field and observed in the region of −8.5–−8.3 ppm. Such a notable displacement of the substituted boron atom clearly indicates the disappearance of the triple C≡N bond and the formation of the double C=N bond. Signals from unsubstituted boron atoms were in the region of −17.5–−18.5 ppm. The amidines obtained were characterised using ^1^H and ^13^C NMR, IR and ESI MS methods. 

In ^1^H NMR spectra, signals from the cationic part were observed as multiplets in the 3.19–3.09, 1.64–1.6, 1.45–1.37 and 1.02–0.97 ppm areas for the tetrabutylammonium cation, and from the hydride hydrogen atoms of the *closo*-dodecaborate anion in the 2.50–0.00 ppm area. The amidine fragment was represented by hydrogen atom signals of the amino group in the region of 9.80–7.60 ppm, the imino group 6.70–5.80 ppm, the acetonitrile methyl group 2.14–2.04 ppm and signals from substituents of the attached amine. The results obtained were in good agreement with previously described data relating to *closo*-decaborate anions. Thus, in the case of derivatives of *closo*-decaborate anions, signals from amino groups were found in a tighter interval of 7.65–6.96 ppm with signals from imino groups occurring in the interval 7.65–6.90 ppm. Therefore, signals from the *exo*-polyhedral group of *closo*-borate anions occurred in the high field area in comparison with *closo*-dodecaborate anion derivatives. 

In ^13^C NMR, the amidine fragment was represented by signals of the amidine carbon atom in the 166.3–162.9 ppm region, and signals of the methyl group of the acetonitrile methyl group in the 21.01–19.3 ppm region signals from substituent of attached amine. For *closo*-decaborate derivatives, signals of amidine carbon atoms were found in the interval 163.3–165.5 ppm. Thus, in this case, cluster type had only a slight influence on the magnitude of the chemical shift.

When nitrilium derivatives transformed into amidine derivatives, triple C≡N bonds turned into double C=N bonds. This process is easily monitored using IR-spectra. The signal from the strong bands of C=N bond vibrations were observed in the range of 1660–1640 cm^−1^. Amino group is represented by group of signals from 3420 to 3250 cm^−1^. This indicates that the NH group participates in the formation of different intra- and intermolecular hydrogen bonds.

X-ray crystallography data provided additional information about the structure of amidine derivatives (Figure 1a–d). Insights into the solid state structure of four amidine derivatives were obtained. Firstly, based on the data obtained, it was established that the addition process of amine led to the formation of product with Z-isomerism. A theoretical investigation was carried out into the reasons for the observed phenomena. In brief, the main driving force is the formation of intramolecular dihydrogen bonds between protons of amino group and hydrogen atoms of boron cluster. Full details are provided in the section devoted to investigating the mechanisms involved. The length of dihydrogen contacts in amidine derivative structures were within the interval of 1.94–2.10 Å. The presence of the intramolecular dihydrogen bond was also demonstrated by conducting a QTAIM analysis of the [B_12_H_11_NCCH_3_NH_2_]^−^ anion in the gas phase. The main electron density parameters of the present hydrogen bond were in good agreement with previously obtained data for deca- and dodecaborate derivatives (see Supporting Information). In theoretical calculations in the gas phase, the value of the bond length (1.66 Å) is shorter than in the case of a crystal structure. A possible explanation for this phenomenon is that, in the solid phase, protons of amino groups form intramolecular, as well as intermolecular, contacts. It is noteworthy that, in the case of [B_12_H_11_NCCH_3_NH_2_]^−^, intermolecular interactions were observed between proton atoms of the amino group and hydrogen atoms of the boron cluster. In the case of 8 and 9 derivatives, there were interactions between the proton atom of the imino group and hydrogen atoms of the boron cluster. The values of these contacts were found in the interval of 2.07–2.18 Å.

Another interesting question concerns the bond orders of the CN bond in the amidine fragment. The lengths of the contacts were quite similar, indicating the presence of a conjugation effect. In addition, the similarity of the CN bond orders was demonstrated using NBO analysis in the case of [B_12_H_11_NCCH_3_NH_2_]^−^. The Wiberg bond order for the interaction between a carbon atom and an imino nitrogen atom is equal to 1.50, and for a carbon atom and an amino nitrogen atom the figure is 1.30. Therefore, both theoretical and experimental data point to the formation of a π-system between two nitrogen atoms and a carbon atom.

Thus, the novel class of borylated amidines of the general form [B_12_H_11_NCCH_3_NHR]^−^, R=H, Alk, Ar was obtained. The preparation of given compounds is based on the addition reaction of amines to nitrilium derivatives of *closo*-dodecaborate anions. This process is characterised by mild synthetic conditions and high yields. One of the process advantages is the possibility to carry out reactions in water solvent without using organic solvents. The proposed approach allows one to obtain borylated amidine with a wide structural diversity and construct a bioinorganic system with biologically active organic amines and amino acids.

### 2.3. Theoretical Investigation of Addition Process Mechanism

The mechanism of the interaction between the nitrilium derivatives of *closo*-dodecaborate anions [B_12_H_11_NCCH_3_]^−^ (Init) and ammonia NH_3_ was theoretically investigated. To consider solvation effects, calculations were performed on tetrahydrofuran C_4_H_8_O and water H_2_O solutions using the SMD model [66] to obtain the main thermodynamic parameters. The formation of the final product [B_12_H_11_NHCCH_3_NH_2_]^−^ includes the following steps (Figure 3).

Firstly, the molecule of ammonia attacked the C-atom of the nitrilium group. Activation barriers of this process were equal to 68 kJ∙mol^−1^ in tetrahydrofuran solution and 65 kJ∙mol^−1^ in water solution (Figure 2 and Figure 3 for water solution; Appendix A for tetrahydrofuran solution). The corresponding transition states of the general form [B_12_H_11_NCCH_3_*NH_3_]^−^ (TS1) have been successfully located on potential energy surfaces. In these transition states, the length of the C≡N bond becomes elongated in comparison to that of the [B_12_H_11_NCCH_3_]^−^. In the initial nitrilium derivatives, the length of the C≡N bond was equal to 1.15 Ǻ in both cases (tetrahydrofuran and water solutions) and in transition states it was equal to 1.17 Ǻ, also in both tetrahydrofuran and water solutions. The distance between the C-atom and the nitrogen atom of ammonia was equal to 2.05 Ǻ in the case of tetrahydrofuran solution and 2.08 Ǻ in the case of water solution. The process of the addition of ammonia to nitrilium derivatives can lead to the formation of two different isomers, E or Z. In practice, however, only the Z-isomer was obtained. Based on the structure data of the transition state, it may be concluded that the driving force that produces Z-isomer is the dihydrogen bonds between the ammonia and *closo*-borate cage. The lengths of hydrogen bonds in the case of tetrahydrofuran solution (2.30 and 2.36 Ǻ) were shorter than in the case of water solution (2.35 and 2.42 Ǻ).

The addition process led to the endergonic formation of [B_12_H_11_NCCH_3_NH_3_]^−^ (Int) as intermediate species. The Gibbs free energy of the addition reaction was 37 kJ∙mol^−1^ (tetrahydrofuran solution) and 25 kJ∙mol^−1^ (water solution). The length of C=N bonds was in the interval 1.24–1.25 Ǻ. The length of the C–NH_3_ bond was equal to 1.52 Ǻ. [B_12_H_11_NCCH_3_NH_3_]^−^ (Int) has one intramolecular non-covalent contact between the hydrogen atom of the NH_3_-group and the H-atom of the cluster cage. In the case of tetrahydrofuran as a solvent, the distance of the intermolecular dihydrogen contact was shorter (1.74 Ǻ) than with the water solution (1.77 Ǻ). 

In the next stage, the H-atom migrates to an additional molecule of ammonia, with the exergonic formation of complexes [B_12_H_11_NCCH_3_NH_2_*NH_4_]^−^ (Comp1). The Gibbs free energy of the H-migration process was −11 kJ∙mol^−1^ with the tetrahydrofuran solution and −15 kJ∙mol^−1^ with the water solution. The activation barrier of this process was extremely low: 1 kJ∙mol^−1^ with tetrahydrofuran solution and 0.6 kJ∙mol^−1^ with water solution. Corresponding transition states (TS2) were localised on the potential energy surface. The main geometry parameters of these transition states have been considered. The length of one N-H bond elongated to 1.19 Ǻ (water solution) and 1.23 Ǻ (tetrahydrofuran solution). The distance between the nitrogen atom of the ammonia and the hydrogen atom of the NH_3_ group from the *exo*-polyhedral substituent was equal to 1.35 Ǻ (water solution) and 1.42 Ǻ (tetrahydrofuran solution). The length of the C-NH_2_ bond was equal to 1.47 Ǻ. The structure of the [B_12_H_11_NCCH_3_NH_2_*NH_4_]^−^ complex was stabilised by several hydrogen bonds. The length of the NH-BH contact was 1.78 Ǻ in the case of tetrahydrofuran solution and 1.87 Ǻ with the water solution.

For the next step, the proton H^+^ migrated from the NH_4_^+^ to amidine N-atom without energy barriers. The exergonic complex [B_12_H_11_NHCCH_3_NH_2_*NH_3_]^−^ (Comp2) was formed. The structure of resulting complex was stabilised by the hydrogen bond between the proton attached to amidine nitrogen atom of the *exo*-polyhedral substituent and the nitrogen atom of ammonia. The length of this contact was 1.89 Ǻ with the water solution and 1.91 Ǻ with the tetrahydrofuran solution. The [B_12_H_11_NHCCH_3_NH_2_*NH_3_]^−^ complex was not stable and exergonically dissociate with formation of [B_12_H_11_NHCCH_3_NH_2_]^−^ (Prod) and NH_3_. The length of the C=N bond was 1.31 Ǻ. The distance between the C-atom and the N-atom from the NH_2_-group was equal to 1.33 Ǻ. The final product had a Z-configuration, which was stabilised by an intermolecular hydrogen bond. The distances for this contact were quite similar in both cases: 2.07 Ǻ and 2.33 Ǻ with tetrahydrofuran solution and 2.07 Ǻ and 2.36 Ǻ with the water solution.

Thus, the interaction mechanism between nitrilium derivatives of *closo*-dodecaborate anions and ammonia has been proposed. The rate-limiting step of this process is the nucleophilic attack of ammonia on the C-atom of the nitrilium group. Based on structure data relating to the transition state, it can be concluded that the main cause of there being only one isomer with a Z configuration was hydrogen bonds between the ammonia and the *closo*-borate cage. Proton migration from one nitrogen atom to another took place with the help of an additional molecule of ammonia. The solvent effects did not play a crucial role, but with water being used for solvent energy, barriers were lower than when tetrahydrofuran solvent was used.

### 2.4. Biological Activity Study

#### 2.4.1. MTT Assay

The toxicity of borylated amidine derivatives was investigated. IC 50 criteria, with the help of MTT assay, was applied. As described above, four different cell lines were selected: NKE (human kidney epithelium), HaCat (human immortalised keratinocytes), U251 (human glioblastoma) and Hep2 (human laryngeal cancer). For BNCT purposes, the boron cluster compound must have the ability to accumulate in cancer cells in a significant quantity. Thus, the first step of the biological property investigation was to assess the viability of cancer cells in the presence of boron cluster anions. The lower the toxicity of these molecular systems concerning the target cell lines, the more the target molecules accumulate in the cancer cells. It is also necessary to compare accumulation in cancer cells and normal cells.

All the compounds studied were low-toxic [62,67,68] and showed no antitumour activity. It This facts is either comparable to, or lower than, the cytotoxicity of BSH used in BNCT and various derivatives based on it [68,69,70]. It is therefore possible, in principle, to achieve therapeutic concentrations of boron atoms in the cell. Thus, the IC50 of the most hydrophilic ammonia addition product is 3–6 times higher than the IC50 of B12-borylated aniline, the most hydrophobic substituent (see Table 1). Borylated ethyl glycinate also has some of the lowest cytotoxicity values for both tumour and non-tumour cell lines.

#### 2.4.2. Complex Formation with BSA and HSA

Albumin is often considered the essential transport protein in the delivery of drugs to tumour cells, so the study of the binding of these proteins to target compounds can be regarded as an essential aspect of developing new substances for BNCT. The binding process of several obtained compounds to BSA (bovine serum albumin) and HSA (human serum albumin) was studied. Binding was evaluated in terms of changes in protein fluorescence intensity upon the sequential addition of substituted dodecaborates (Figure 4 and Figure 5). Tryptophan, located in the protein’s hydrophobic cavity, is involved in binding guest molecules [71]. In this regard, compounds containing hydrophobic and incredibly aromatic functional groups form the most stable complexes.

As previously established, fluorescence quenching allows one to approximately estimate the binding parameters of *closo*-borates to transport proteins [72]. The data obtained on the binding constants correlated with the degree of hydrophobicity of the compounds under study. Thus, for the substituted aniline, the dissociation constant of the complex with HSA was 2.07*10^5^ M^−1^, which almost exactly coincides with data previously obtained for similar amidine based on the *closo*-decaborate anion, indicating that the cluster geometry had little effect on the stability of non-covalent complexes with albumin. At the same time, the absence of an aryl group in the amidine substituent reduced the binding constant by a factor of approximately 10 (See Table 2).

## 3. Materials and Methods

IR spectra of the compounds were recorded on an Infralum FT-08 IR Fourier spectrophotometer (NPF Lumex AP) in the region 4000–400 cm^−1^ with a resolution of 1 cm^−1^. Samples were prepared as KBr pellets. 

^1^H, ^13^C and ^11^B[^1^H] NMR spectra of solutions of the studied substances in CD_3_CN or CD_2_Cl_2_ were recorded on a Bruker MSL-300 pulsed Fourier spectrometer (Ettlingen, Germany) at frequencies of 300.3, 75.49 and 96.32 MHz, respectively, with internal deuterium stabilisation. Tetramethylsilane or boron trifluoride ether was used as the external standard, respectively.

ESI mass spectra The LC system consisted of two LC-20AD pumps (Shimadzu, Japan) and autosampler was coupled online with an LCMS-IT-TOF mass spectrometer equipped with an electrospray ionisation source (Shimadzu, Japan). The HRMS spectra were acquired in direct injection mode without column. Mass spectra were obtained in the m/z range from 120 to 700 Da (for negative ionisation mode) and 100–700 for positive mode. Other MS parameters: Detector Voltage: 1.55 kV. Nebulizing Gas: 1.50 L/min. CDL Temperature: 200.0 °C CDL. Heat Block Temperature: 200.0 °C. ESI Voltage: 4.50 kV. Event Time: 300 ms. Repeat: 3. Ion Accumulation: 30 ms. Instrument tuning (mass calibration and sensitivity check) was carried out before analysis.

Computational details. The full geometry optimisation of all model structures has been carried out at the ωB97X-D3/6-31++G(d,*p*) level of theory with the help of the ORCA 4.2.1 program package [73] (the atom-pairwise dispersion correction with the zero-damping scheme was utilised [74]. The convergence tolerances for the geometry optimization procedure were: energy change = 5.0 × 10^−6^ Eh, maximal gradient = 3.0 × 10^−4^ Eh/Bohr, RMS gradient = 1.0 × 10^−4^ Eh/Bohr, maximal displacement = 4.0 × 10^−3^ Bohr, and RMS displacement = 2.0 × 10^−3^ Bohr. The spin restricted approximation for the model structures with closed electron shells has been employed. Symmetry operations have not been applied during the geometry optimisation procedures for all model structures. The Hessian matrices have been calculated numerically for all optimised model structures in order to prove the location of correct minima on the potential energy surfaces (no imaginary frequencies for all reactants, intermediates and final products; only one imaginary frequency for transition states). The connectivity of each reaction step was also confirmed using the intrinsic reaction coordinate (IRC) calculation from the transition states [75,76,77]. The solvent effects were considered using the Solvation Model based on Density (SMD) [66]. The Cartesian atomic coordinates for all optimised equilibrium model structures are presented in Supporting Information as xyz-files.

X-ray diffraction experiments were conducted at the Center for collective use of physical methods of research of the Frumkin Institute RAS on an automatic four-circle diffractometer with a Bruker KAPPA APEX II area detector [78] (radiation MoKα) at 20 °C. The unit cell parameters were refined on all the data [79]. The structure was solved by the direct method [80] and refined by full-matrix least-squares on *F^2^* for all data in the anisotropic approximation for all non-hydrogen atoms [81]. H atoms of a borohydride cluster were located from the difference Fourier map and refined with isotropic temperature factors equal to 1.2 *U*_eq_ (B). H atoms of the organic part of the structure were placed in geometrically calculated positions with isotropic temperature factors, equal to 1.2 *U*_eq_ (C) for CH_2_ groups, and 1.5 *U*_eq_ (C) for the CH_3_ ones. The absolute structure was not determined. The atomic coordinates are deposited with the Cambridge Crystallographic Data Centre (Deposition Numbers 2122676-2122679). Images are created using the OLEX2 package [82].

Tetraphenylphosphonium salts for X-ray structure analysis were prepared by adding the equimolar value of (PPh_4_)Cl to the solution corresponding amidine in the minimal volume of dichloromethane.

Solvents of reagent and special purity grades, as well as amines Sigma-Aldrich and Panreac (99.7%), were used without any additional purification.

### 3.1. Synthesis of (NBu_4_)[B_12_H_11_(NCCH_3_)]

A glass pressure vessel (for the experimental setup see the Appendix A) was charged with 626 mg (1 mmol) (NBu_4_)_2_[B_12_H_12_], 20 mL CH_3_CN, 0.3 mL CF_3_COOH. Heating was performed on an oil bath placed on a magnetic plate. The oil bath was heated to a temperature of 150 °C and held for 30 min. The solution was cooled down to room temperature for 90 min, then the solution was concentrated using a rotary evaporator. The concentrated solution was diluted with 20 mL of glacial acetic acid and filtered through a Schott glass filter. The precipitate was washed with 50 mL of glacial acetic acid and 50 mL of diethyl ether, then dried over P_2_O_5_ in a desiccator.

(NBu_4_)[B_12_H_11_(NCCH_3_)] 2

Yield: 76%. ^11^B[^1^H]-NMR (CD_2_Cl_2_) δ (ppm): −13.5 (s, 1B, *B*-N), −16.3 (s, 11B, *B*-H). ^1^H NMR (CD_2_Cl_2_) δ (ppm): 2.5–0.0 (m, 11H, B-H), 3.15 (8H, NBu_4_), 2.61 (s, 3H, C-CH_3_), 1.61 (8H, NBu_4_), 1.45 (8H, NBu_4_), 1.01 (12H, NBu_4_). ^13^C NMR (CD_2_Cl_2_) δ (ppm): 108.9 (NC-CH_3_), 59.4 (NBu_4_), 24.4 (NBu_4_), 20.2 (NBu_4_), 13.9 (NBu_4_), 4.5 (NC-CH_3_). IR (CH_2_Cl_2_, cm^−1^, selected bands): 2500 ν (B-H), 2352 ν (B-N). MS (ESI) m/z: 182.2369 (A refers to the molecular weight of [B_12_H_11_(NCCH_3_)], calculated for {[A]-} 182.2316).

(NBu_4_)[B_12_H_11_(NHC(NH_2_)CH_3_)] 3a

An aqueous ammonia solution was added to the solution of 1. (106 mg 0.25 mmol) in C_2_H_5_OH. The reaction mixture was refluxed for two hours. The product was filtered and dried over P_2_O_5_ in a desiccator.

Alternative method: to the solution of 1 (106 mg 0.25 mmol) in THF was added THF ammonia solution. The product was filtered and dried in a vacuum.

(NBu_4_)[B_12_H_11_(NHC(NH_2_)CH_3_)] Yield: 94%. ^11^B[^1^H]-NMR (CD_3_CN) δ (ppm): −8.5 (s, 1B, *B*-N), −17.5 (s, 10B, *B*-H), −18.8(s,1B, *B*-H). ^1^H NMR (CD_3_CN) δ (ppm): 2.5–0.0 (m, 11H, B-H), 7.59 (s, 1H, =C-NH_2_), 6.60 (s, 2H, B=NH, =C-NH_2_), 3.09 (8H, NBu_4_), 2.04 (s, 3H, =C-CH_3_), 1.61 (8H, NBu_4_), 1.37 (8H, NBu_4_), 0.96 (12H, NBu_4_). ^13^C NMR (CD_3_CN) δ (ppm): 166.3 (NH=C-CH_3_), 59.3 (NBu_4_), 24.3 (NBu_4_), 21.0 (NH=C-Me), 20.3 (NBu_4_), 13.8 (NBu_4_). IR (CH_2_Cl_2_, cm^−1^, selected bands): 3432, 3365, 3345, 3297, 3244 ν (N-H), 2483 ν (B-H), 1662 ν (C=N). MS (ESI) m/z: 199.2606 (A refers to the molecular weight of [B_12_H_11_(NHC(NH_2_)CH_3_)], calculated for {[A]-} 199.2581).

(NBu_4_)[B_12_H_11_(NHC(NHCH_3_)CH_3_)] 3b

To the solution of 1 (106 mg 0.25 mmol) in C_2_H_5_OH was added an aqueous solution of CH_3_NH_2_. The reaction mixture was refluxed for two hours. The product was filtered and dried over P_2_O_5_ in a desiccator.

(NBu_4_)[B_12_H_11_(NHC(NHCH_3_)CH_3_)] Yield: 88%. ^11^B[^1^H]-NMR (CD_2_Cl_2_) δ (ppm): −8.5 (s, 1B, *B*-N), −17.3 (s, 10B, *B*-H), −18.7 (s,1B, *B*-H). ^1^H NMR (CD_2_Cl_2_) δ (ppm): 2.5–0.0 (m, 11H, B-H), 7.97 (s, 1H, =C-NH-NH_3_), 5.98 (s, 1H, B=NH), 3.14 (8H, NBu_4_), 2.94 (d, 3H, NH-CH_3_, J = 5.1 Hz,), 2.09 (s, 3H, =C-CH_3_), 1.62 (8H, NBu_4_), 1.44 (8H, NBu_4_), 1.02 (12H, NBu_4_). ^13^C NMR (CD_2_Cl_2_) δ (ppm): (CD_2_Cl_2_) δ (ppm): 165.4 (NH=C-CH_3_), 59.4 (NBu_4_), 30.4 (NH-CH_3_), 24.4 (NBu_4_), 20.2 (NH=C-CH_3_), 20.1 (NBu_4_), 13.9 (NBu_4_). IR (CH_2_Cl_2_, cm^−1^, selected bands): 3403, 3342, 3273 ν (N-H), 2490 ν (B-H), 1653 ν (C=N). MS (ESI) m/z: 213.2755 (A refers to the molecular weight of [B_12_H_11_(NHC(NHCH_3_)CH_3_)], calculated for {[A]-} 213.2738).

(NBu_4_)[B_12_H_11_(NHC(NHCH_2_COOC_2_H_5_)CH_3_)] 3c

Triethylamine (1.1 mmol; 0.150 mL) was added to a 10 mL aqueous solution of the glycine ethyl ester hydrochloride (2.0 mmol), and the resulting mixture was stirred for 15 min at room temperature, then CH_2_Cl_2_ (15 mL) was added. The organic phase was separated, dried over CaCl_2_, and poured into a 1 (106 mg 0.25 mmol) solution in CH_2_Cl_2_ (5 mL). The mixture was stirred at room temperature for 10 min in a dry argon atmosphere. H_2_O (15 mL) was added, and the pH value of the reaction mixture was adjusted to 2–3 with 1 M aqueous HCl. The organic phase was separated, dried over CaCl_2_, and evaporated on a rotary evaporator. The product was recrystallised from C_2_H_5_OH.

(NBu_4_)[B_12_H_11_(NHC(NHCH_2_COOC_2_H_5_)CH_3_)] Yield: 73%. ^11^B[^1^H]-NMR (CD_2_Cl_2_) δ (ppm): −8.5 (s, 1B, *B*-N), −17.2 (s,11B, *B*-H). ^1^H NMR (CD_2_Cl_2_) δ (ppm): 2.5–0.0 (m, 11H, B-H), 8.39 (s, 1H, =C-NH-), 6.27 (s, 1H, B=NH-), 4.25 (q, 2H, -CH_2_-CH_3_, J = 7.2 Hz), 4.02 (d, 2H, NH-CH_2_-COO, J = 6 Hz), 3.15 (8H, NBu_4_), 2.10 (s, 3H, =C-CH_3_), 1.63 (8H, NBu_4_), 1.45 (8H, NBu_4_), 1.30 (q, 3H, -CH_2_-CH_3_, J = 7.1 Hz), 1.02 (12H, NBu_4_). ^13^C NMR (CD_2_Cl_2_) δ (ppm): 168.2 (-COOEt), 165.2 (NH=C-CH_3_), 62.7 (-CH_2_-CH_3_), 59.3 (NBu_4_), 45.4 (NH-CH_2_-COO), 24.4 (NBu_4_), 20.5 (NH=C-CH_3_), 20.1 (NBu_4_), 14.4 (-CH_2_-CH_3_), 13.9 (NBu_4_). IR (CH_2_Cl_2_, cm^−1^, selected bands): 3372, 3318, 3264 ν (N-H), 2488 ν (B-H), 1743 ν (C=O), 1650 ν (C=N). MS (ESI) m/z: 285.2959 (A refers to the molecular weight of [B_12_H_11_(NHC(NHCH_2_COOC_2_H_5_)CH_3_)], calculated for {[A]-} 285.2949).

(NBu_4_)[B_12_H_11_(NHC(NHC_3_H_7_)CH_3_)] 3d

To the solution of 1 (106 mg 0.25 mmol) in CH_2_Cl_2_ (5 mL) was added 0.5 mmol C_3_H_7_NH_2_. The reaction mixture was stirred for 15 min at room temperature. H2O (5 mL) was added, and the pH value of the reaction mixture was adjusted to 2–3 with 1 M aqueous HCl. The organic phase was separated, dried over CaCl_2_, and evaporated on a rotary evaporator. The product was recrystallised from C_2_H_5_OH.

(NBu_4_)[B_12_H_11_(NHC(NHC_3_H_7_)CH_3_)] Yield: 83%. ^11^B[^1^H]-NMR (CD_2_Cl_2_) δ (ppm): −8.5 (s, 1B, *B*-N), −17.3 (s, 10B, *B*-H), −18.4 (s,1B, *B*-H). ^1^H NMR (CD_2_Cl_2_) δ (ppm): 2.5–0.0 (m, 11H, B-H), 8.01 (s, 1H, =C-NH-), 5.84 (s, 1H B=NH), 3.66 (m, 1H, NH-CH), 3.15 (8H, NBu_4_), 2.10 (s, 3H, =C-CH_3_), 1.64 (8H, NBu_4_), 1.43 (8H, NBu_4_), 1.27 (d, 6H, CH(Me)_2_, J = 6.5 Hz), 1.02 (12H, NBu_4_). ^13^C NMR (CD_2_Cl_2_) δ (ppm): 163.0 (NH=C-CH_3_), 59.3 (NBu_4_), 46.9 (NH-CH), 24.4 (NBu_4_), 23,4 (CH(CH_3_)_2_), 20.1 (NH=C-CH_3_), 19.8 (NBu_4_), 13.9 (NBu_4_). IR (CH_2_Cl_2_, cm^−1^, selected bands): 3420, 3355, 3319, 3254 ν (N-H), 2488 ν (B-H), 1642 ν (C=N). MS (ESI) m/z: 241.3068 (A refers to the molecular weight of [B_12_H_11_(NHC(NHC_3_H_7_)CH_3_)], calculated for {[A]-} 241.3051).

(NBu_4_)[B_12_H_11_(NHC(NHC_4_H_9_)CH_3_)] 3e

The method of obtaining is similar to 3d.(NBu_4_)[B_12_H_11_(NHC(NHC_4_H_9_)CH_3_)] Yield: 91%. ^11^B[^1^H]-NMR (CD_2_Cl_2_) δ (ppm): −8.5 (s, 1B, *B*-N), −17.3 (s, 10B, *B*-H), −18.7 (s,1B, *B*-H). (CD_2_Cl_2_) δ (ppm): 2.5–0.0 (m, 11H, B-H), 8.01 (s, 1H, =C-NH-), 5.91 (s, 1H B=NH), 3.66 (dd, 2H, NH-CH_2_, J = 7.1 Hz.), 3.15 (8H, NBu_4_), 2.09 (s, 3H, =C-CH_3_), 1.64 (10H, NBu_4_, CH_2_-CH_2_-), 1.43 (10H, NBu_4_ CH_2_-CH_3_), 1.01 (12H, NBu_4_), 0.95 (t, 3H CH_2_ -CH_3_, J = 7.3 Hz.). ^13^C NMR (CD_2_Cl_2_) δ (ppm): 164.3 (NH=C-CH_3_), 59.3 (NBu_4_), 44.0 (NH-CH_2_-CH_2_), 31.8 (-CH_2_-CH_2_), 24.4 (NBu_4_), 20.2 (NH=C-CH_3_), 20.1 (NBu_4_), 20.1 (CH_2_-CH_3_), 13.9 (CH_2_-CH_3_), 13.8 (NBu_4_). IR (CH_2_Cl_2_, cm^−1^, selected bands): 3403, 3330, 3264 ν (N-H), 2486 ν (B-H), 1647 ν (C=N). MS (ESI) m/z: 255.3219 (A refers to the molecular weight of [B_12_H_11_(NHC(NHC_4_H_9_)CH_3_)], calculated for {[A]-} 255.3207).

(NBu_4_)[B_12_H_11_(NHC(NH(CH_2_)_2_OH)CH_3_)] 3f

The method of obtaining is similar to 3d. (NBu_4_)[B_12_H_11_(NHC(NH(CH_2_)_2_OH)CH_3_] Yield: 77%. ^11^B[^1^H]-NMR (CD_2_Cl_2_) δ (ppm): −8.5 (s, 1B, *B*-N), −17.3 (s, 10B, *B*-H), −18.5 (s,1B, *B*-H). ^1^H NMR CD_2_Cl_2_) δ (ppm): 2.5–0.0 (m, 11H, B-H), 8.18 (s, 1H, =C-NH-), 6.01(s, 1H B=NH), 3.66 (t, 2H, NH-CH_2_-CH_2_-OH, J = 5.1 Hz.), 3.40 (dt, 2H, NH-CH_2_-CH_2_-OH, J = 5.5 Hz.), 3.15 (8H, NBu_4_), 2.14 (s, 3H, =C-CH_3_), 1.62 (8H, NBu_4_), 1.43 (8H, NBu_4_), 1.01 (12H, NBu_4_). ^13^C NMR CD_2_Cl_2_) δ (ppm): 165.2 (NH=C-CH_3_), 61.5 (CH_2_-OH), 59.3 (NBu_4_), 46.3 NH-CH2 (NH-CH_2_), 24. 3 (NBu_4_), 20.4 (NH=C-CH_3_), 20.1 (NBu_4_), 13.8 (NBu_4_). IR (CH_2_Cl_2_, cm^−1^, selected bands): 3529 ν (O-H), 3402, 3327, 3266 ν (N-H), 2488 ν (B-H), ν (C=N). MS (ESI) m/z: 243.2857 (A refers to the molecular weight of [B_12_H_11_(NHC(NHC_2_H_4_OH)CH_3_)], calculated for {[A]-} 243.2843).

(NBu_4_)[B_12_H_11_(NHC(NHC_6_H_5_)CH_3_)] 3g

The method of obtaining is similar to 3d

(NBu_4_)[B_12_H_11_(NHC(NHC_6_H_5_)CH_3_)] Yield: 84%. ^11^B[^1^H]-NMR (CD_2_Cl_2_) δ (ppm): −8.3 (s, 1B, *B*-N), −17.0 (s, 11B, *B*-H). ^1^H NMR (CD_2_Cl_2_) δ (ppm): 2.5–0.0 (m, 11H, B-H), 9.80 (s, 1H, =C-NH-), 7.50–7.20 (m, 5H, C_6_H_5_), 6.39(s, 1H B=NH), 3.16 (8H, NBu_4_), 2.09 (s, 3H, =C-CH_3_), 1.62 (8H, NBu_4_), 1.44 (8H, NBu_4_), 1.00 (12H, NBu_4_).

^13^C NMR (CD_2_Cl_2_) δ (ppm): 164.4 (NH=C-CH_3_), 136.4, 130.1, 128.4 (C_6_H_5_), 59.4 (NBu_4_), 24. 5 (NBu_4_), 20.8 (NH=C-CH_3_), 20.2 (NBu_4_), 13.9 (NBu_4_). IR (CH_2_Cl_2_, cm^−1^, selected bands): 3391 3340, 3299, 3248 ν (N-H), 2490 ν (B-H), 1641 ν (C=N). MS (ESI) m/z: 275.2900 (A refers to the molecular weight of [B_12_H_11_(NHC(NHC_6_H_5_)CH_3_)], calculated for {[A]-} 275.2894).

(NBu_4_)[B_12_H_11_(NHC(NHC_6_H_11_)CH_3_)] 3h

The method of obtaining is similar to 3d.

(NBu_4_)[B_12_H_11_(NHC(NHC_6_H_11_)CH_3_)] Yield: 94%. ^11^B[^1^H]-NMR (CD_2_Cl_2_) δ (ppm): −8.6 (1B, *B*-N), −17.3 (s, 10B, *B*-H), −18.6 (s,1B, *B*-H). ^1^H NMR (CD_2_Cl_2_) δ (ppm): 2.5–0.0 (m, 11H, B-H), 8.06 (s, 1H, C-NH), 5.82 (s, 1H, C=NH), 3.37 (м 1H NH-CH), 3.15 (8H, NBu_4_), 2.14 (s, 3H, C-CH_3_), 1.95–1.20 (m 10H CH(C_2_H_4_)_2_CH_2_), 1.64 (8H, NBu_4_), 1.43 (8H, NBu_4_), 1.01 (12H, NBu_4_). ^13^C NMR (CD_2_Cl_2_) δ (ppm): 162.9 (NH=C-CH_3_), 59.1 (NBu_4_), 53.1 (NH-CH), 33.4 (CH(CH_2_CH_2_)_2_CH_2_), 33.4 (CH(CH_2_CH_2_)_2_CH_2_), 24.4 (CH(CH_2_CH_2_)_2_CH_2_), 24.2 (NBu_4_), 20.0 (NBu_4_), 19.3 (NH=C-CH_3_), 13.8 (NBu_4_). IR (CH_2_Cl_2_, cm^−1^, selected bands): 3404, 3355, 3320, 3251 ν (N-H), 2488 ν (B-H), ν (C=N). MS (ESI) m/z: 281.3377 (A refers to the molecular weight of [B_12_H_11_(NHC(NHC_6_H_11_)CH_3_)], calculated for {[A]-} 281.3364).

### 3.2. Synthesis of Sodium Salts for Biological Assays

To the solution of the corresponding amidine derivative in the form of tetrabutylammonium salt (0.200 mmol) in the dichloromethane (5 mL) was added the solution of Na[BPh_4_] (0.195 mmol) in the water (5 mL). The emulsion was stirred for 2 h at room temperature. Then, the aqueous layer was separated, filtered and concentrated on a rotary evaporator.

### 3.3. Biological Study

#### 3.3.1. Cell Cultures

All cell lines NKE (human kidney epithelium), HaCat (human immortalised keratinocytes), U251 (human glioblastoma) and Hep2 (human laryngeal cancer) were grown in DMEM medium supplemented with 10% foetal calf serum (FCS), and pen-strep 50 units/mL (all from PanEco, Moscow, Russian Federation).

#### 3.3.2. MTT Assay

To investigate the cytotoxic effect during the experiment, cells were seeded in 96-well plates (BD Micro-FinePlus, Franklin Lakes, NJ, USA) at 4 × 10^3^ cells/well in 180 μL of culture medium. After 24 h, compounds were added at a final volume of 20 μL/well and maintained for 72 h at 37 °C in 5% CO_2_. After this time, 10 μL of MTT reagent solution (5 mg/mL, PanEco, Moscow, Russian Federation) was added to each well and left for another 3.5 h. Formazan formed in cells was dissolved in 100 μL of dimethyl sulfoxide (DMSO, PanEco, Russia). The optical density of the solution was measured using a multiwell spectrophotometer MultiScan MCC 340 (Labsystems, Kennett Square, PA, USA) at 540 nm. The experiment was repeated at least three times for each compound. Stock and serial dilutions of the compounds were made in deionised water on the experiment day. The concentration of compounds giving 50% of the maximum toxic effect (IC50) was calculated from titration curves. The results were graphing, and statistical processing was performed using the Excel program package (Microsoft, Redmond, WA, USA).

#### 3.3.3. Fluorescence Quenching Studies

The fluorescence spectra were obtained on the Fluorat-02 Panorama fluorescence spectrophotometer (Lumex, Saint-Peterburg, Russia) in quartz cell (1 cm × 1 cm). To study the quenching of the protein fluorescence by the *closo*-borates, the fluorescence spectra of the analyzed protein solution with a concentration of 0.2 mg/mL in the 0.05 M Tris-HCl aqueous buffer solution with pH 7.9 and those after the successive addition of aqueous solutions of the borylated amidines were registered. The fluorescence was excited at 280 nm, and its intensity value was registered at λmax of a protein under study. The fluorescence quenching experiments were performed at room temperature. The concentration of amidines was changed from 0, 5, 10, 15, 25 to 50 μM.

## 4. Conclusions

In the present work, a combined approach to the synthesis and property investigation of novel borylated amidines based on *closo*-dodecaborate anions with the general form [B_12_H_11_NCCH_3_NHR]^−^, R=H, Alk, Ar was carried out. The synthetic strategy for the synthesis of given derivatives had two stages. In the first stage, nitrilium derivatives [B_12_H_11_NCCH_3_]^−^ were obtained. The method for the preparation of nitrilium derivative was substantially improved. In the second stage, goal products were obtained by applying amines addition reaction to nitrilium derivatives of the *closo*-dodecaborate anion. This process was characterised by mild synthetic conditions and high yields of target products. The mechanism of interaction between nitrilium derivatives of *closo*-dodecaborate anions and ammonia has been proposed. The rate-limiting step of the addition process in the nucleophilic attack of ammonia on the C-atom of nitrilium group which was proved on the base of quantum chemistry calculation. Based on structure data relating to the transition state and X-ray crystallography, it is possible to conclude that the main cause of only one isomer with a Z configuration being obtained was hydrogen bonds between ammonia and the *closo*-borate cage. Moreover, it was observed that the resulting *closo*-decaborate amidines were low toxic and could form labile complexes with transport peptides such as albumin.

## Data Availability

The data presented in this study are available in Appendix A and from the authors.

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
