# Peer review of "Primary Amine Nucleophilic Addition to Nitrilium *Closo*-Dodecaborate [B_12_H_11_NCCH_3_]^−^: A Simple and Effective Route to the New BNCT Drug Design"

_ijms, 2021, doi:10.3390/ijms222413391_

Round 1

Reviewer 1 Report

In their manuscript, the authors report on the synthesis of closo-dodecaborate amidines and their properties.

More specifically, the nitrilium ion [B12H11NCCH3]– is used as a starting material and converted to amidines by addition reactions with amines. A series of new compounds is presented. Characterization is carried out using spectroscopic methods and X-ray crystallography. Furthermore, the authors investigate the mechanism of nucleophile addition computationally. Finally, they also report on bioactivity, presenting MTT assay data and binding to bovine serum albumin and human 352 serum albumin.

Overall, this is a very nice study on the functionalization of the closo-dodecaborate cluster and potential applications of the derivatives obtained. The primary data are in agreement with the conclusions made by the authors.

Given the increasing interest in boron cluster compounds and their applications in the life sciences, the results are of interest to a broad audience.

Recommendation in International Journal of Molecular Sciences is recommended after minor revisions.

In a revised submission, please address the following issues:

1) The following publications on the advances in dodecaborate functionalization should be discussed in the intruduction and cited appropriately:

- Chem. Soc. Rev. 2013, 42, 3318.

- Pure Appl. Chem. 2018, 90, 733.

- 'Recent Advances in the Selective Functionalization of Anionic Icosahedral Boranes and Carboranes'

Book chapter in 'Synthetic Inorganic Chemistry'2021, Elsevier, ISBN: 9780128184295, 343-389

2) closo-dodecaborate amidines have already been reported in the following publication:

Molecules 2018, 23, 3137.

Add this publication to the references and discuss it in the introduction.

3) The improved synthesis of the nitrilium ion [B12H11NCCH3]– is an important part of the manuscript. The authors state that the reaction temperature and time are crucial in order to obtain a high yield and purity of this compound. An autoclave is used for heating the reaction mixture to 150 °C.

In the supporting information file, please explain in detail the experimental setup. Show pictures of the autoclave used and how the reaction mixture was heated.

Additionally, please explain the heating time: Was it 30 minutes after reaching 150 °C? Or 30 minutes total, including the heating ramp? How did you cool the mixture down to room temperature? Over how many minutes?

These details are very valuable for all researchers who want to reproduce this synthesis.

Author Response

We graciously thank the reviewer for bringing those works to our attention. The citations have been incorporated on page 2.

In addition, we have expanded the description of nitrilium derivative [B12H11NCCH3]– synthesis and added photos of equipment to the supporting info.

Thank you for your time and consideration.

Reviewer 2 Report

I their manuscript entitled “Primary Amine Nucleophilic Addition to Nitrilium closo-Do-decaborate [B12H11NCCH3]: A Simple and Effective Route to the New BNCT Drug Design” Andrey P. Zhdanov and coworkers present an interesting combined synthetic and biochemical study on boron cluster chemistry. 

  The synthesis of the acetonitrile adduct has been improved and some selected addition reactions to the activated NC unit is described. This part of the work is nice and has been performed thoroughly but is close to routine. The nice combination of this synthetic work with a toxicity study and a binding study with BSA and HSA makes the work very interesting and thus, justifies publication.

The manuscript represents a valuable addition to closo-B12 chemistry and should be published in the International Journal of Molecular Sciences.

Author Response

We thank this reviewer for his or her kind words and for taking the time to review our manuscript.